# Chemical Characterization and Enantioselective Analysis of *Tagetes filifolia* Lag. Essential Oil and Crude Extract

**DOI:** 10.3390/plants13141921

**Published:** 2024-07-12

**Authors:** Vladimir Morocho, Anghela Chamba, Paulo Pozo, Mayra Montalván, Alírica I. Suárez

**Affiliations:** 1Departamento de Química, Universidad Técnica Particular de Loja (UTPL), Loja 1101608, Ecuador; msmontalvan@utpl.edu.ec; 2Carrera de Bioquímica y Farmacia, Universidad Técnica Particular de Loja (UTPL), Loja 1101608, Ecuador; anchamba@utpl.edu.ec (A.C.); papozo4@utpl.edu.ec (P.P.); 3Facultad de Farmacia, Universidad Central de Venezuela, Caracas 1040, Venezuela

**Keywords:** essential oil, anethole, *Tagetes filifolia* Lag.

## Abstract

The essential oil (EO) of *Tagetes filifolia* Lag. was obtained from dried plant material through Clevenger-type steam distillation and analyzed using gas chromatography–mass spectrometry (GC/MS), a gas chromatography–flame ionization detector (GC/FID) and enantioselective gas chromatography. The results showed 50 compounds (93.33%) with a predominance of oxygenated monoterpenes. The main components were trans-anethole (55.57 ± 9.83%), tridecene <1-> (8.66 ± 0.01), methyl chavicol (5.81 ± 0.85%) and Neophytadiene (3.45 ± 0.88) Enantioselective analysis revealed linalool and <methyl-γ-> ionone as enantiomers. The identification of secondary metabolites from the ethyl acetate extract obtained by maceration was performed by GC-MS, NMR and by a literature comparison, determining the presence of mostly trans-anethole and a mixture of two triterpenes, fernenol and lupeol.

## 1. Introduction

Medicinal properties have been attributed to approximately 10,000 plant species worldwide [1]. Within this vast botanical diversity, more than 3000 species, including representatives from the Asteraceae, Lamiaceae and Apiaceae families, are recognized for their ability to produce essential oils [2].

*Tagetes* is an endemic American genus, comprising approximately 55 species that inhabit tropical and subtropical regions, ranging from the southwestern United States to Argentina. The *Tagetes* genus is distinguished by its diverse flower morphologies and colors, which span a spectrum from yellowish to reddish, accompanied by a characteristic aroma [3,4]. In addition to their aromatic qualities, plants within the *Tagetes* genus are also known for their rich secondary metabolite content, which includes thiophenes, phenols, flavonoids, coumarins and terpenes [4].

There are over 23,000 species in the Asteraceae family, which includes a large number of flowering plants. Some of the most commonly used species are chamomile (*Matricaria recutita* L.), echinacea (*Echinacea purpurea* L.), yarrow (*Achillea millefolium* L.) and wormwood (*Artesmisa* spp.). The most striking and remarkable aspect of the Asteraceae family is the arrangement of their flowers in tightly packed heads. Antibacterial, antifungal and herbicidal activities are observed in the EOs and extracts of these species [5].

*Tagetes filifolia* Lag., commonly known as Sacha anis, Anisillo, Allpa anís, Panpa anís, Sacha anís, Anís com$\overset{´}{u}$n, Anís de campo, Anís de monte, Anisillo or Anis serrano, is an herbaceous plant with a rich history of traditional medicine and culinary practices. It has been utilized as a cooking ingredient, a remedy for coughs, an antipsychotic agent, for alleviating menstrual and digestive cramps and as infusion to combat dysentery [6,7,8]. Additionally, the oil extracted from this plant is frequently employed as an airway decongestant [9].

*T. filifolia* is renowned for its bioactive compounds, including trans-anethole, allyl anisole and β-caryophyllene, which have demonstrated efficacy as repellents and inhibitors against the reproduction and growth of fungi, bacteria, mites, nematodes, Diptera, stored grains weevils, lice and aphids [10].

The present research aims to conduct a phytochemical analysis of the ethyl acetate extract obtained from the leaves of *T. filifolia* Lag. and to identify the chemical composition and enantiomeric distribution of its essential oil.

## 2. Results

The essential oil of the aerial parts of *Tagetes filifolia* Lag. was obtained by steam distillation for 4 h. As it is an analytical distillation method, the yield of the essential oil was calculated based on the internal standard milligrams, determining a percentage yield of 0.03%. The chemical composition, enantiomeric analysis and isolations of compounds are discussed below.

### 2.1. Chemical Composition of EO from T. filifolia

The results of the chemical analysis of essential oil were obtained by GC/MS and GC/FID using a DB-5 MS non-polar column, and are shown in Table 1.

A total of fifty compounds were found, representing 93.33% of the essential oil. Figure 1 shows the gas chromatogram of essential oil from *T. filifolia*.

Anethole (55.57%), tridecene <1-> (8.66%) and methyl chavicol (5.81%) were the most representative compounds in the essential oil. Other minor compounds present (≤5%) included Cymene <2,5-dimethoxy-p-> (2.3%), ionone <methyl-γ->, germacrene D (1.67%), bisabolene (1.82%) <β-> and (-)-spathulenol (1.68%). The analysis revealed that the principal chemical group was oxygenated monoterpenes, representing 61.72% of the analyzed essential oil.

### 2.2. Enantioselective Analysis of the Essential Oil of Tagetes filifolia Lag.

A GC column coated with 2,3-diacethyl-6-tert-butylsilyl-β-cyclodextrin as a chiral selector was used for the enantioselective analysis. The analysis revealed linalool and ionone <methyl-γ-> as enantiomers with enantiomeric excess (e.e.) of 13.62% and 81.1%, respectively. The results of the enantioselective analysis are shown in Table 2.

### 2.3. Characterization of Compounds

Trans-anethole (**1**) and a mixture of fernenol (**2**) and lupeol (**3**) were isolated from the total ethyl acetate extract, obtaining a yield of 4.20%. These compounds were characterized by spectroscopic techniques such as MS and NMR (1H and 13C) in one- and 2-dimensional experiments and were compared with data from the literature [13,14,15]. The NMR data of the identified compounds are shown in Figure 2.

In the GC-MS analysis, trans-anethole was identified based on the calculated retention index according to the methodology of Van Den Dool and Kratz, and their mass spectrum, which were same as those in the study by Adams [11]. Finally, the presence of this compound was corroborated using NMR. In the chromatogram, the compound and MS spectra are shown in Figure 3 and Figure 4.

The mixture of fernenol (**2**) and lupeol (**3**) was eluted in isocratic condition with hexane/dichloromethane (7:3), and NMR data corroborated the literature [13,14,15].

## 3. Discussion

In the chemical analysis of *T. filifolia* essential oil, trans-anethole (55.57 ± 0.83%), tridecene <1-> (8.66 ± 0.01) and methyl chavicol (5.81 ± 0.85%) were identified as the main components. Previous research in different locations, such as Peru, Ecuador and Venezuela, corroborated the predominant presence of trans-anethole and methyl chavicol, with percentages ranging from 68.2% to 87.5% and 10.7% to 19.7%, respectively [16,17,18,19], although in Italy, a different chemical composition was reported with methyl chavicol as the majority compound (90.4%) [20]. The presence of anethole is infrequent, although in the genus *Tagetes*, the species *T. filifolia* and *T. mandonii* contain anethole in their essential oil. In *T. mandonii*, 12.7% of the oil corresponds to phenylpropanoids and only 9.2% corresponds to cis-anethole [21].

Despite the differences, the predominance of oxygenated monoterpenes supports the traditional use of this plant as a flavoring and condiment [19]. Compared to other species of the genus *Tagetes*, *T. filifolia* shares *trans*-anethole as the main component with *T. pusilla*, while distinct species show distinct chemical compositions, such as piperitone, (*E*)-tagetone and terpinolene. These variations are attributed to environmental and genetic factors [22,23,24].

Enantiomeric analysis is important in the case of EOs, whose main property is their aroma, as they present different olfactory properties and aromas due to their various enantiomeric compositions [25]; the enantioselective analysis of the essential oil of *T. filifolia* revealed the separation of the enantiomer’s linalool and ionone <methyl-γ-> with an enantiomeric excess of 13.62% and 81.1%, respectively. However, enantioselective analyses in *Tagetes* species have not been reported previously. Linalool (3,7-dimethyl-1,6-octadien-3-ol) is a widely known chiral monoterpene that is present in hundreds of botanical species, and is a monoterpene whose contribution to flavor and fragrance is dependent not only on concentration but also on the dominant enantiomer form. (S)-(+)-linalool and (R)-(−)-linalool are the dominant forms in clary sage, lavender, hops and hemp. On the other hand, (S)-(+)-linalool is the dominant form in cardamom, coriander and catnip. Aromatherapeutic benefits from linalool are becoming increasingly recognized in consumer products, such as the soothing effects arising from such botanical inclusions as lavender or clary sage. (R)-(−)-linalool has been reported to cause feelings of calmness in its consumers. Linalool exhibited antidepressant-like effects in the TST and FST in mice, possibly through the modulation of serotonergic and noradrenergic systems [26,27]. Three compounds were identified in the plant extract of *T. filifolia* leaves, the first one being the main component, named Trans-anethole (1), which is an ether-type unsaturated aromatic monoterpene with the molecular formula C_10_H_12_O and molecular weight 148.2 g/mol.

*Trans*-anethole is a notable compound among the oils that make up anise and fennel, mainly because it gives them their characteristic aroma. Studies attribute oxidative activity to it, as the free radicals that make up the phenolic side chain conjugated to the structural aromatic ring can conjugate with the radical cation, leading to electron delocalization in the aromatic ring and stabilization in the 1–4 interaction of the methoxy group [28]. On the other hand, it also possesses antifungal activity [29], as it has been described to inhibit *Candida* species by inducing oxidative stress in the opportunistic fungal pathogen through mitochondrial death cascades, since the hydrophobic and lipophilic structures of the essential oil interact with the fungal membrane, resulting in the alteration of its fluidity and properties, as described in Ref. [28]. In addition, a potential gastroprotective effect has been determined [30].

In view of the scarce bibliography, the identification of fernenol (**2**) and lupeol (**3**), both triterpenes with the molecular formula C_30_H_50_O and molecular weight 426.7 g/mol, determined their presence in plant material but with the peculiarity of their potential in different pharmacological fields. Fernenol has been shown in previous studies to be effective in inhibiting the tumor growth cycle and in inducing apoptosis in tumor cells both in vivo and in vitro [31]. On the other hand, lupeol has shown great biological and pharmacological potential both in vivo and in vitro because it is very beneficial in treating diabetes, renal and hepatic toxicity and heart disease, and is also very effective as an anti-arthritic, anti-inflammatory and anticancer agent, reducing the risk of breast, colon and prostate cancer by 20%, as it induces cell apoptosis and inhibits cell proliferation, migration and invasion [32].

In addition, lupeol has been reported as an antineuroinflammatory, antioxidative, antimicrobial and skin-protective agent, as studies have shown that it enhances wound healing by stimulating keratinocyte migration and increasing the contraction of fibroblasts embedded in a collagen matrix [33].

## 4. Materials and Methods

### 4.1. General Information

The chemical analyses of essential oil were carried out with a gas chromatograph (Trace 1310), coupled to simple quadrupole mass spectrometry detector, model ISQ 7000 (Thermo Fisher Scientific, Waltham, MA, USA). Additionally, a common flame ionization detector (FID) complemented the same instrument. The mass spectrometer was operated in SCAN mode (scan range 40–350 *m/z*), with the electron ionization (EI) source set at 70 eV. A non-polar column based on 5%-phenyl (equiv) Polysilphenylene-siloxane was applied to the qualitative and quantitative analyses. The non-polar column was TR-5MS (30 m long, 0.25 mm internal diameter, and 0.25 μm film thickness), purchased from Thermo Fisher Scientific, Waltham, MA, USA. The enantioselective analysis was carried out through an enantioselective capillary column, based on 2,3-diacetyl-6-tert-butyldimethylsilyl-β-cyclodextrin as a chiral selector (25 m × 250 μm internal diameter × 0.25 μm phase thickness), and purchased from Mega, Milan, Italy. GC purity grade helium, from Indura, Guayaquil, Ecuador, was used as the carrier gas, set at the constant flow of 1 mL/min. For all the GC analyses, the analytical-purity-grade solvents, the mixture of *n*-alkanes (C_9_–C25) and the internal standard (n-nonane) were purchased from Sigma-Aldrich (St. Louis, MO, USA). Silica gel 60 (Merck KGaA, Darmstadt, Germany, from 0.063 to 0.200 mm) was used as the stationary phase for column chromatography. All NMR data experiments were carried out with a 500 MHz Bruker spectrometer (Bruker, Billerica, MA, USA), and CDCl3 deuterated solvent was purchased from Sigma-Aldrich (St. Louis, MO, USA).

### 4.2. Plant Material

*Tagetes filifolia* Lag. was collected during its flowering period in May 2022 at Cerro Villonaco, located in Loja province, Ecuador, with coordinates 3°59′47″ S. The plant material was deposited in the Herbarium of the Universidad Técnica Particular de Loja. The leaves were collected and dried at 35 °C for 48 h. This investigation was carried out with the permission of the Ministry of Environment, Water, and Ecological Transition of Ecuador, with MAATE registry number MAE-DNB-CM-2016-0048.

### 4.3. Essential Oil Extraction

The dried plant material (65 g) was distillated using a Clevenger apparatus with a solution (2.0 mL) containing n-nonane as the internal standard (7 mg) in cyclohexane (10 mL). This process was carried out over 4 h. The resulting cyclohexane solution was directly injected into the GC instrument.

### 4.4. Qualitative and Quantitative Analyses

The qualitative and qualitative analysis of *T. filifolia* EO was carried out with a gas chromatograph model Trace 1310 gas coupled to a simple quadrupole mass spectrometry detector, model ISQ 7000 (Thermo Fisher Scientific, Walthan, MA, USA). A non-polar column DB-5ms (0.25 mm × 30 m with a thickness of 0.25 µm) with 5%-phenyl-methylpolysiloxane was used as the stationary phase.

The chromatography temperature gradient was from 60 °C to 250 °C. The first oven temperature was 60 °C, and it was then increased 2 °C/min to 100 °C, 3 °C/min to 150° C, 5 °C/min to 200 °C and 15 °C/min to reach the final temperature of 250 °C, which was maintained for 5 min until the end. The carrier gas was Helium at a constant flow rate of 1.0 mL/min. Mass spectra were taken at 70 eV. The range of scan mass was from 40 *m/z* to 350 *m/z*. The duration of the chromatographic run was 60 min.

To analyze the essential oil obtained from the dried plant, we injected 1 μL of essential oil using a 1:80 split injection mode. Before analyzing the samples, alkanes from C9 to C22 were injected under the same conditions to calculate the Retention Index (RI). The software used for the data process was Chromeleon 7. The identification of compounds was carried out to compare the RI and mass spectra with the information available in the literature [11,12]. The percentage of each compound of the oils was computed using the normalization method from the GC peak areas, calculated by means of three injections from each oil, without using correction factors.

### 4.5. Enantioselective Analysis

The enantioselective analysis was carried out using GC-MS based on a 2,3-diacetyl-6-tert-butyldimethylsilyl-β-cyclodextrin capillary column as a chiral selector. The temperature of the injector port was 200 °C with a constant pressure of 75 kPa. The oven temperature was programmed to 65−200 °C. The duration of the chromatographic run was 96 min. The conditions of sample analyses and the alkanes were the same as those described in quantitative and qualitative analyses.

Enantiomeric excess was determined by subtracting the minor enantiomer from the major enantiomer, expressed as a percentage enantiomer from the major enantiomer, and expressed as a percentage. The retention index obtained was compared with information available in the literature [25].

### 4.6. Extraction and Isolation of Secondary Metabolites

The dried leaves of *T. filifolia* (400 g) were exhaustively submitted to solvent extraction by maceration at room temperature with ethyl acetate (EtOAc) for 24 h. This process was performed three times. The obtained solutions were filtered and concentrated under reduced pressure for solvent removal.

The EtOAc extract (2 g) was submitted to silica column chromatography, with an extract/silica ratio of 1:100. The column was eluted according to a gradient of increasing polarity, from hexane-ethyl acetate 90:10 to 100% ethyl acetate, producing a total of 11 fractions (PP01-12/09). The fractions were combined according to their TLC patterns, and eluted with n-hexane/EtOAc according to metabolite polarity. Fraction PP01/09 (400mg g) was eluted with an isocratic polarity of 100% hexane, to obtain compound **1** (52.8 mg), resulting in an aromatic unsaturated ether-type monoterpene known as trans-anethole identified by NMR and gas chromatography coupled to mass spectrometry (GC-MS).

Fraction PP03/09/15 (225 mgg) was obtained with polarity isocratic hexane/dichloromethane (7:3) to produce compounds **2** and **3** (48.2 mg), identified a mixture of triterpenes known as fernenol and lupeol, also identified by NMR.

## 5. Conclusions

The leaves of *Tagetes filifolia* Kunth contain an essential oil with a distillation yield of 0.03%, predominantly composed of monoterpenes, which account for more than 62% of the chemical composition. The hydrodistillation of the leaves produces an essential oil, with the major compounds identified as anethole (55.57%), tridecene <1-> (8.66%) and methyl chavicol (5.81%). The identification of anethole and two triterpenes, among other compounds, as the major constituents of *T. filifolia* is consistent with reports from other species of this genus.

## Figures and Tables

**Figure 1 plants-13-01921-f001:**
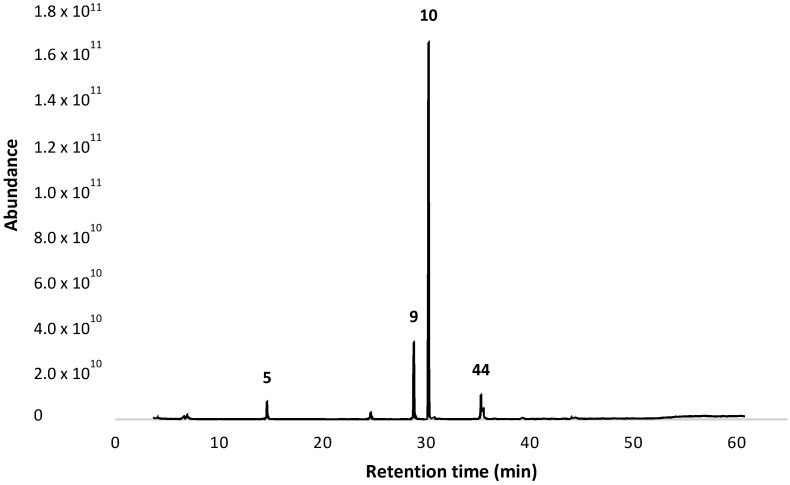
Typical gas chromatogram of the essential oil from *Tagetes filifolia* through a 5% phenyl-methylpolysiloxane capillary column.

**Figure 2 plants-13-01921-f002:**
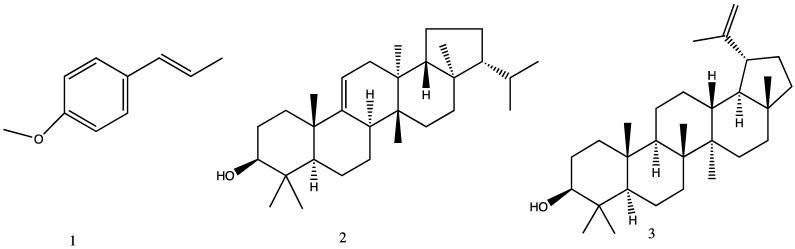
Structure of compounds (**1**–**3**) isolated from *Tagetes filifolia*.

**Figure 3 plants-13-01921-f003:**
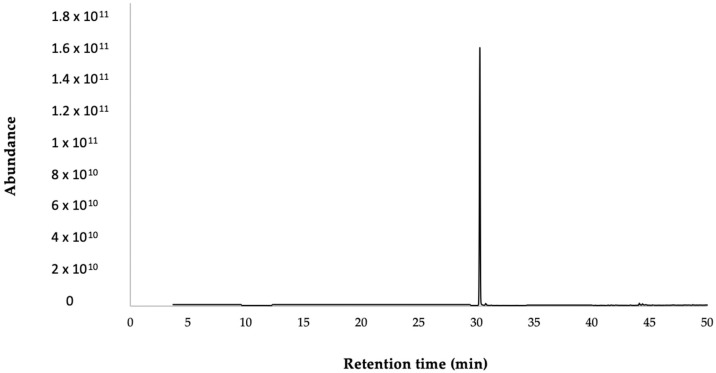
Typical chromatogram of *trans*-anethole.

**Figure 4 plants-13-01921-f004:**
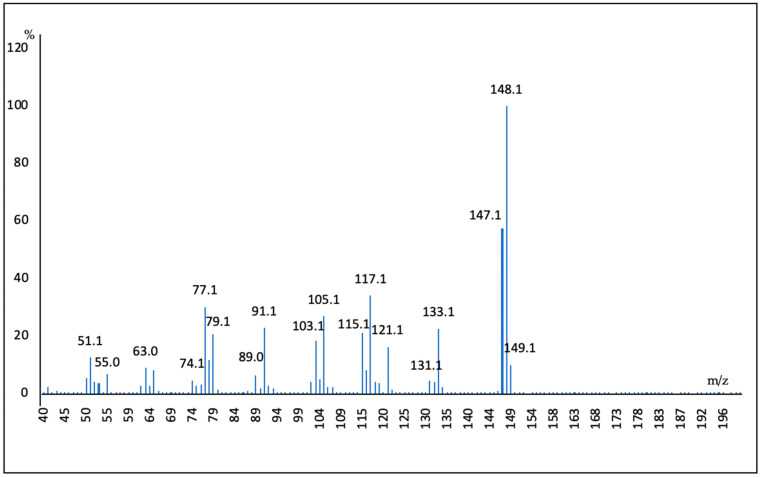
Mass spectrum of *trans*-anethole.

**Table 1 plants-13-01921-t001:** Chemical composition of the essential oil from *T. filifolia* Lag.

			5% Phenyl-Methylpolysiloxane				
N°	Compounds ^a^	RI ^b^	RI ^c^	% *	Type	CF	MM (Da)
1	Carene <δ-3->	1011	1008	0.15 ± 0.08	MH	C_10_H_16_	200.32
2	Cymene <ρ->	1029	1020	0.08 ± 0.1	MH	C_10_H_14_	134.11
3	Cyclohexanedione <3-methyl-1,2->	1092	1085	0.01 ± 0.03	OC	C_7_H_10_O_2_	126.07
4	Linalool	1106	1105	0.21 ± 0.26	MHO	C_10_H_18_O	154.14
5	Methyl chavicol	1209	1195	5.81 ± 0.85	MHO	C_10_H_12_O	148.08
6	Thymol, methyl ether	1239	1232	0.14 ± 0.05	OC	C_11_H_16_O	164.12
7	Cis-Anethole	1266	1282	0.13 ± 0.06	MHO	C_10_H_12_O	148.08
8	Acetanisole <ο->	1275	1291	0.26 ± 0.27	OC	C_9_H_10_O_2_	150.07
9	Tridecene <1->	1294	1290	8.66 ± 0.01	OC	C_13_H_26_	182.20
10	*Trans*-Anethole	1300	1282	55.57 ± 0.83	MHO	C_10_H_12_O	148.08
11	Octanediol <1,8->	1331	1339	0.06 ± 0.04	OC	C_8_H_18_O_2_	146.13
12	Copaene <α->	1380	1374	0.11 ± 0.02	SH	C_15_H_24_	204.19
13	Modheph-2-ene	1386	1382	0.26 ± 0.04	SH	C_15_H_24_	204.19
14	Isocomene <α->	1393	1387	0.19 ± 0.03	SH	C_15_H_24_	204.19
15	Methyl eugenol	1415	1403	0.38 ± 0.07	OC	C_11_H_14_O_2_	204.19
16	Cymene <2,5-dimethoxy-p->	1423	1424	2.03 ± 0.11	OC	C_12_H_18_O_2_	194.13
17	Bergamotene <α-trans->	1437	1432	0.08 ± 0.01	SH	C_15_H_24_	204.19
18	Elemene <γ->	1447	1434	0.08 ± 0.01	SH	C_15_H_24_	204.19
19	Farnesene <(E)-β-	1457	1454	0.10 ± 0.08	SH	C_15_H_24_	204.19
20	Ionone <methyl-γ->	1484	1480	1.67 ± 0.19	OC	C_14_H_22_O	206.16
21	Germacrene D	1488	1480	1.54 ± 0.32	SH	C_15_H_24_	204.19
22	Bicyclogermacrene	1504	1500	0.28 ± 0.02	SH	C_15_H_24_	204.19
23	Farnesene < (E,E)-α->	1510	1505	0.53 ± 0.14	SH	C_15_H_24_	204.19
24	Bisabolene <β->	1515	1505	1.82 ± 0.31	SH	C_15_H_24_	204.19
25	Zonarene	1528	1528	0.18 ± 0.10	SH	C_15_H_24_	204.19
26	Cameroonan-7-α-ol	1535	1510	0.02 ± 0.04	SHO	C_15_H_26_O	222.20
27	Nerolidol <(E)->	1571	1561	0.43 ± 0.02	SH0	C_15_H_26_O	222.20
28	Ionone <dimethyl->	1578	1565	0.19 ± 0.03	SHO	C_15_H_24_O	220.19
29	(-)-Spathulenol	1592	1577	1.68 ± 0.11	SHO	C_15_H_24_O	220.19
30	Oplopenone <β->	1596	1607	0.24 ± 0.01	SHO	C_15_H_24_O	220.19
31	Copaen-4-α-ol <β->	1600	1590	0.20 ± 0.05	SHO	C_15_H_24_O	220.19
32	Ledol	1608	1602	0.24 ± 0.02	SHO	C_15_H_26_O	222.20
33	Humulene epoxide II	1621	1608	0.24 ± 0.02	SHO	C_15_H_24_O	220.19
34	Isobornyl isobutanoate <6-hydroxy->	1638	1643	0.34 ± 0.03	OC	C_14_H_24_O_3_	240.17
35	Vulgarone B	1642	1651	0.17 ± 0.01	SHO	C_15_H_22_O	218.16
36	Allohimachalol	1644	1661	0.17 ± 0.02	SHO	C_15_H_24_O	220.19
37	Cadinol <epi-α->	1660	1638	0.44 ± 0.17	SHO	C_15_H_26_O	222.20
38	Cadinol <α->	1674	1673	0.5 ± 0.13	SHO	C_15_H_26_O	222.20
39	Khusinol	1688	1680	0.03 ± 0.01	SHO	C_15_H_24_O	220.19
40	Propyl chromone <2->	1702	1706	0.58 ± 0.05	OC	C_12_H_12_O_2_	188.08
41	Humulene <14-hydroxy-α->	1718	1713	0.3 ± 0.02	SHO	C_15_H_24_O	220.19
42	Longifolol <iso->	1726	1728	0.14 ± 0.03	SHO	C_15_H_26_O	222.20
43	Amorpha-4,9-diene <7,14-anhydro->	1763	1755	0.11 ± 0.02	SHO	C_15_H_22_O	218.16
44	Neophytadiene	1841	1836	3.45 ± 0.88	OC	C_20_H_38_	278.30
45	2-Pentadecanone,6,10,14-trimethyl-	1852	1847	0.32 ± 0.04	OC	C_18_H_36_O	268.28
46	Hexadecanoic acid, methyl ester	1938	1926	0.10 ± 0.02	OC	C_17_H_34_O_2_	270.26
47	Catalponol <epi->	1968	1988	0.67 ± 0.07	SHO	C_15_H_18_O_2_	230.13
48	Pseudo phytol <(6Z,10Z)->	1977	1988	0.44 ± 0.07	OC	C_20_H_36_O	292.28
49	Palmitic Acid, TMS derivative	2048	2047	0.09 ± 0.05	OC	C_19_H_40_O_2_	300.30
50	Methyl linoleate	2107	2095	0.27 ± 0.22	OC	C_19_H_34_O_2_	294.25
	Monoterpenes Hydrocarbonated			0.23			
	Monoterpenes Oxygenated			61.72			
	Sesquiterpenes Hydrocarbonated			6.81			
	Sesquiterpenes Oxygenated			5.77			
	Others			18.8			
	Total			93.33			

* X ± SD (standard deviation); ^a^ compounds listed in order of elution; ^b^ calculated retention index measured relative to n-alkanes (C_9_–C_22_); ^c^ literature retention index [11,12], CF: condensed formula; MM: monoisotopic mass

**Table 2 plants-13-01921-t002:** Enantioselective analysis of the essential oils of *T. filifolia* Lag.

N.	Enantiomers	2,3-Diethyl-6-*tert*-Butyldimethylsilyl-β-Cyclodextrin		
		RI ^a^	Enantiomeric Distribution (%)	Enantiomeric Excess (%)
1	(R)-(−)-linalool	1317	56.81	13.62
2	(S)-(+)-linalool	1319	43.19	
3	Ionone <methyl-γ->	1601	90.55	81.10
4	Ionone <methyl-γ->	1608	9.45	

^a^ RI (calculated retention index).

## Data Availability

The original contributions presented in the study are included in the article, further inquiries can be directed to the corresponding author.

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
