# Peer review of "Chemical Characterization and Enantioselective Analysis of Tagetes filifolia Lag. Essential Oil and Crude Extract"

_plants, 2024, doi:10.3390/plants13141921_

Round 1
Reviewer 1 Report
Comments and Suggestions for Authors
This manuscript tried to reported the phytochemical composition from essential oil and solvent extract. However, the description of the experimental method mentioned in the article has defects and some important information were missing, which may not make the experimental results with high credibility and persuasiveness.
1. In Section “4. Materials and Methods”, what is the extraction yield of the essential oil, as well as the ethyl acetate extract?
2. There is not the manufacturer information of the columns of DB-5ms or enantioselective analysis. What are the software and database used in the identification of the volatile?
3. It is mentioned that the internal standard was added, but what is the Quantitative Analyses method?
4. In Section “4.5 Extraction and isolation of secondary metabolites”, the isolation procedure is unclear and there are many mistakes.
5. Only 3 compounds were isolated and purified from this plant. Are these three components the major phytochemical constituents and what are their contents?
Comments on the Quality of English LanguageExtensive editing of English language required.
Author Response
- Comments: In Section “4. Materials and Methods”, what is the extraction yield of the essential oil, as well as the ethyl acetate extract?
Response: The methodology has been modified according to surgeries
- . Comments: There is not the manufacturer information of the columns of DB-5ms or enantioselective analysis. What are the software and database used in the identification of the volatile?
Response: In Materials and Methods, in the General information item, the required information has been entered.
- . Comments: It is mentioned that the internal standard was added, but what is the Quantitative Analyses method?
Response: In item 4.3. Qualitative and Quantitative Analyses modifications have been made.
- .Comments: In Section “4.5 Extraction and isolation of secondary metabolites”, the isolation procedure is unclear and there are many mistakes.
Response: The suggested changes were made
- Comments: Only 3 compounds were isolated and purified from this plant. Are these three components the major phytochemical constituents and what are their contents?
Response: The modification was made, and the performance of the compounds was added.

Reviewer 2 Report
Comments and Suggestions for Authors
The manuscript investigates the chemical composition and enantiomeric distribution of Tagetes filifolia Lag. essential oil and ethyl acetate extract and the isolation of trans-anethole. After reading this work, I have some observations:
1. I suggest rereading the text for typos and mistakes.
2. Rewrite the title to reflect the extract as well.
3. The introduction is too brief.
4. References appear in two styles. Please follow the journal’s instructions for citations.
5. Lines 46-48: provide references to support this statement.
6. Figure 1 seems unnecessary. Consider removing it.
7. Table 2: It should be “.” rather than “,” in the % column.
8. Lines 101, 107, and 119: avoid using references in the results section. This section should focus on describing the findings of the current study. Rewrite or relocate these parts.
9. Figure 5: The compound numbers do not match.
10. What was the yield of steam distillation?
11. What solvent was used to dilute the oil for analysis?
12. Line 232: EtOAc
13. Add a schematic diagram summarizing the separation of the EtOAc extract.
14. Rewrite the conclusions more concisely for clarity. Focus on the major findings.
Comments on the Quality of English LanguageThere are some typos and grammatical mistakes.
Author Response
- Comments: I suggest rereading the text for typos and mistakes.
Response: The suggested corrections were made
- Comments : Rewrite the title to reflect the extract as well.
Response: The title has been modified: Chemical characterization and enantioselective analysis of Tagetes filifolia Lag. essential oil and crude extract
- Comments: The introduction is too brief.
Response:. Added text in the introduction
- Comments : References appear in two styles. Please follow the journal’s instructions for citations.
Response: The manuscripts have been modified
- Comments: Lines 46-48: provide references to support this statement.
Response: Corrections were made
- Comments : Figure 1 seems unnecessary. Consider removing it.
Response: The chromatogram is modified by adding more information about the compounds.
- Comments: Table 2: It should be “.” rather than “,” in the % column.
Response: Corrections were made
- Comments: Lines 101, 107, and 119: avoid using references in the results section. This section should focus on describing the findings of the current study. Rewrite or relocate these parts.
Response: These parts were rewritten
- Comments: Figure 5: The compound numbers do not match.
Response: Chemical structures were reorganized
- Comments: What was the yield of steam distillation?
Response: The percentage yield of the essential oil and extract was added
- Comments: What solvent was used to dilute the oil for analysis?
Response: The methodology was modified
- Comments: Line 232: EtOAc
Response: Corrections were made
- Comments: Add a schematic diagram summarizing the separation of the EtOAc extract.
Response: The methodology for obtaining the total extract was modified
- Comments: Rewrite the conclusions more concisely for clarity. Focus on the major findings.
Response: The conclusions were rewritten

Reviewer 3 Report
Comments and Suggestions for Authors
NMR data of compounds 1,2 and 3 it is well known; tables 3 and 4 are not necessary.
All reference numbers should be placed in square brackets [ ].
In line 52, change filolia by filifolia
Please discuss the relevance enantioselective analysis of the essential oil of Tagetes filifolia Lag.
Figure 2 doesn't show the NMR data of the identified compounds. Please remove figure 2. Lines 89-90
Author Response
- Comments: NMR data of compounds 1,2 and 3 it is well known; tables 3 and 4 are not necessary.
Response: NMR tables were removed
- Comments: All reference numbers should be placed in square brackets [ ].
Response: References were corrected
- Comments: In line 52, change filolia by filifolia
Response: This species was corrected
- Comments: Please discuss the relevance enantioselective analysis of the essential oil of Tagetes filifoliaLag.
Response: In discussions the importance of enantiomeric analysis was added
- Comments: Figure 2 doesn't show the NMR data of the identified compounds. Please remove figure 2. Lines 89-90
Response: The figures were modified

Round 2
Reviewer 1 Report
Comments and Suggestions for Authors
The manuscript has been revised as suggested.
Comments on the Quality of English LanguageMinor editing of English language required.
Reviewer 2 Report
Comments and Suggestions for Authors
The authors have made substantial improvements in the revised manuscript. I think it is ready for publication.
Reviewer 3 Report
Comments and Suggestions for Authors
The authors responded appropriately to the suggestions.